# Entrepreneurship, Sport, Sustainability and Integration: A Business Model in the Low-Season Tourism Sector

**Ricardo Reier Forradellas [1],\*, Sergio Náñez Alonso [1], Javier Jorge Vázquez [1], Miguel Ángel Echarte Fernández [1] and Nicolas Vidal Miró [2]**

[1] School of Economics and Business, Catholic University of Avila, 05005 Ávila, Spain; sergio.nanez@ucavila.es (S.N.A.); javier.jorge@ucavila.es (J.J.V.); mangel.echarte@ucavila.es (M.Á.E.F.)
[2] Consell of Mallorca, 07010 Palma, Spain; nicolasvidalmiro@gmail.com
\* Correspondence: ricardo.reier@ucavila.es

**Abstract:** The global tourism reality is changing, and not only because of the COVID-19 pandemic. This reality is especially representative in countries such as Spain, which are highly dependent on the income generated by the tourism sector. In these destinations, it is necessary to seek innovation and specialization in the sector in order to achieve new business models. This need is even more pressing in destinations overcrowded by the sun and beach effect, as is the case of Mallorca. The proposed work combines the concepts of sports tourism with the development of a wealth-generating business model that will contribute to promoting a tourism that is sustainable, environmentally friendly and deseasonalized. On the other hand, the proposed work will contribute to promoting integration and equality in the participation of women in sports through the development of a model based on the promotion of women's football. Using the methodology of case analysis, the results of all the approaches outlined are provided, and we obtained a wealth-generation model that is easily replicable and sustainable over time. This work provides a solution to the combination of a sustainable business model that links responsible tourism, the promotion of women's sport and the generation of wealth.

**Keywords:** entrepreneurship; women's sports development; sustainable tourism; deseasonalized tourism; economic development; case analysis methodology

## 1. Introduction

In this first section we give an overview of the objectives of the work and the way it is structured. The general objective is the development of an entrepreneurial model that generates wealth based on a sustainable and deseasonalized tourism model. As such, this business model would be carried out through a novel proposal for sustainable tourism, such as the development of women's sport (in this case women's football). In this way, it will be possible to combine aspects of integration and equality in the field of sport through business plans that are sustainable themselves, beyond traditional subsidies or public sponsorship.

The first aspect to explain is the specific choice of Mallorca as a location. It is important to keep in mind that the United Nations Educational, Scientific and Cultural Organization (UNESCO) itself considers tourism to be the most important industry in the world, ahead of industries such as the automobile and chemical industries. This fact is even more relevant in the case of Spain, where tourism (in pre-COVID-19 times) represented 12.3% of the global GDP in 2019, rising to 42.1% of the GDP in the case of the Balearic Islands (Figuerola 2020).

Authors such as Santoyo Calderón and García Ramos (among others) seem to agree that the current challenge for established tourism models is to balance the quantity with the quality of tourism, attracting tourism with greater added value to ensure

the economic, social and environmental sustainability of the model in the long term (Santoyo Calderón 2015). Logically, as can be seen in Table 1, it is in those areas that are the most dependent on tourism as an economic activity (mainly the Canary Islands and the Balearic Islands in the case of Spain) where these actions are most necessary.

**Table 1.** Tourism as a share of GDP, 2019**. Millions of Euros.**

|  | **Gross Tourism Production** | **Tourism GDP** | **% Tourism GDP** |
| --- | --- | --- | --- |
| Andalucía | 35,827 | 19,347 | 12.46 |
| Aragón | 5,931 | 3,203 | 8.88 |
| Asturias | 4,369 | 2,359 | 10.39 |
| Baleares | 23,326 | 12,596 | 42.12 |
| Canarias | 24,946 | 13,471 | 30.46 |
| Cantabria | 1,936 | 1,046 | 7.99 |
| Castilla León | 9,27 | 5,006 | 8.77 |
| Castilla la Mancha | 5,679 | 3,066 | 7.66 |
| Cataluña | 48,204 | 26,03 | 11.64 |
| Valencia | 24,497 | 13,228 | 12.16 |
| Extremadura | 2,904 | 1,568 | 12.16 |
| Galicia | 11,181 | 6,037 | 8.46 |
| Madrid | 35,354 | 19,091 | 9.92 |
| Murcia | 5,37 | 2,9 | 9.55 |
| Navarra | 3,673 | 1,984 | 10 |
| País Vasco | 9,771 | 5,276 | 7.35 |
| La Rioja | 1,293 | 698 | 8.57 |
| Ceuta/Melilla | 698 | 377 | 11.99 |
| **Nacional Total** | **254,19** | **137,262** | **11.8** |

Source: Our own elaboration based on the work of Figuerola M. 2020.

In general, according to various sources (López 2019; Milano 2017), it seems clear that tourism is destined to be the fundamental pillar and support point in the evolution of the economy of traditional destinations (such as the case of the island of Mallorca), but it urgently needs to modernize its growth models. This same idea is reflected in the Spanish Tourism Plan Horizon 2020 drawn up by the Ministry of Industry, Tourism and Trade, which specifically includes the search for sustainable alternatives to traditional sun and beach destinations (data collected from the Spanish Ministry of Industry, Tourism and Sport 2017).

Once the field of action has been defined, it is necessary to explain the reasons as to why women's football was chosen as the driving force for the development of the proposed model. Women's sports in Spain are on the rise. Over the last few years, the achievements made by Spanish female athletes, both individually and as a group, have not stopped increasing the impact of women's sports on society. In the last Olympic Games, played in Rio de Janeiro in 2016, women were key for Spain to move forward in the medal standings. Athletes such as Mireia Belmonte (swimming), Marina Alabau (windsurfing), Ángela Pumariega (sailing), Carolina Marín (badminton), Maialen Chorreaut (canoeing) or athletes on the national teams, such as basketball, handball and synchronized swimming, supported a sport weight traditionally reserved for men. The increasing visibility of women athletes and women's sports in the media is helping Spanish society become more aware of the difference in treatment that has existed, and still exists, between men's and women's sports; because of this increasing visibility, society is becoming more and more aware of this reality, regardless of gender. (Serra

et al. 2018; Anderson and White 2017). This social repercussion is encouraging more women to get interested and to start doing sports as well (Nicolson 2017).

Similarly, the progress of women's sports also represents progress in terms of equality and integration (Leruite Cabrera et al. 2015). The administrations themselves have implemented different regulations focused on the sports field for the sake of gender equality (Robles and Escobar 2007). Many authors have considered an academic perspective with regard to the situation of women in the field of sports and perspectives and needs for strengthening this relationship (Puig and Soler 2003; Puig and Soler 2004).

On the other hand, tourism has become a fundamental factor in the dissemination of culture and, in addition, one of the most relevant economic supports as a generator of employment and wealth, even more so in a country like Spain (Lara de Vicente and López Guzman 2004). Spain, as seen in this paper, is one of the most sought-after tourist destinations in Europe (the scope of the study focuses on a typical profile of tourists from Europe), and the Balearic Islands and Mallorca are one of its greatest exponents. However, tourism promoters and managers have always sought to evolve through the specialization and diversification of tourism services and products to respond to innovative demands that both generate wealth and go beyond the traditional sun and beach tourist destination (Nogueras Carrasco 2010). Mallorca, as a tourist destination, has the necessary and appropriate infrastructures for this business model to develop positively, by combining an ideal climate for the proposed project with all the services offered as one of the preferred destinations for European tourists, the ultimate target of this line of business. The idea of this study is to unite the promotion of women's sports, soccer in this case, not through public initiative, but through sustainable business models with the participation of various actors. This research aims to obtain results that will help to better understand and develop not only the social and cultural dimension of women's sports, but also the business dimension and its subsequent media and economic impact. Women, young people and the elderly are the groups that have experienced the highest growth rates in terms of sports. Entities and companies linked to the sports field do well to heed the latest studies on sports practice that indicate two interesting and revealing aspects: the emergence of new business opportunities linked to sports, with sports tourism being one of its main references, and the increase in spending that occurs in relation to these new practices (Murillo 2019). As seen below, the methodology carried out in this study consists of a case analysis methodology, widely used in the field of social sciences. Through this methodology, this paper will develop a sustainable and easily replicable business model applied to the development of women's sport.

Choosing women's football (soccer) as the basis for this study was no accident. Although it is undeniable women's position in the world of sports is increasingly entrenched, it is especially striking that women's football has become a booming sport. To prove this statement, just check the successes of the Spanish women's soccer teams in recent years:

- Classification for the first time in history for a World Cup (Canada);
- U17 European Champions;
- European Under-19 runners-up.

At the club level, since 2015 two of the women's soccer league teams have participated in the Union of European Football Associations (UEFA) Women's Champions League (FC Barcelona and Atlético de Madrid). In terms of sports licenses, although the percentage of female representation is still very low compared to men, the Royal Spanish Football Federation occupies fifth place in the total number of federations in Spain in terms of the number of female licenses, with a total of 44,873. Analyzing the historical evolution, women's football licenses have doubled in the last 10 years, from 21,396 licenses in 2007 to 44,873 licenses in 2018.

As seen in the present investigation, the existing potential to relate a traditional and highly demanded tourist destination, such as the island of Mallorca, especially at the European level, and the development of a sustainable business sector linked to both the city and women's sports—in our case, the sport of football—serve as a theoretical framework for this study and as a methodological basis for possible similar studies.

The starting point to justify this entrepreneurial project is the current situation regarding the practice of women's sports in Spain and the evolution of women's soccer in recent years. Taken together, the importance of sports tourism as a generator of wealth within a sustainable and deseasonalized tourism model helps to complement the traditional sun and beach tourism that is characteristic of Spanish tourism, especially in a context such as the Balearic Islands and Mallorca, where this model of mass tourism has generated more and more problems (Jacob et al. 2003; Orfila-Sintes et al. 2005). These novel approaches are even more necessary in contexts such as the current one, marked by the coronavirus pandemic, with dramatic effects on the tourism sector.

Subsequently, the proposed business model is explained through an exhaustive study of the business model, in which the structure and organization of the services provided are analyzed, as well as the overall impact of these actions. In this last part, a brief economic-financial breakdown is developed to study the viability of the business line, its suitability to the suggested environment and the desired factors of sustainability and integration (Garin-Munoz and Montero-Martín 2007; Fortuny et al. 2008).

There are currently companies active in Spain that provide similar services, although it is true that their location tends to focus on the Mediterranean coastline (Costa del Sol and Costa Blanca), prioritizing men's soccer. Therefore, the provision of this service in the Balearic Islands—specifically in Mallorca—beyond the business model presented and the possibility of creating wealth, also serves to improve the sustainability of tourism as a whole with regard to the environment. It is important to point out that the high season for this business model coincides with the low season for tourism on the island, that is to say, it takes place outside the summer months. This situation also allows us to offer a service of the highest quality at much more affordable prices than those that would occur in high season. This issue is also highly advantageous for one of the negative effects of the low season for the island's economy: the consequent lowering of prices of hotel establishments to have an advantage over other destinations. This model takes advantage of the favorable climate and the quality of the establishments, together with business profitability and a plan, which could serve to launch the same business in several points of the island. Therefore, the activity presented can be replicated in other parts of the Balearic Islands.

Thus, in this context, we provide a solution to the issues raised at the beginning of this work under a business entrepreneurship project focused on the development of an inclusive, sustainable and deseasonalized business model, focused on the promotion and development of women's soccer as an example of wealth-generating sports tourism. It proposes the creation of an entrepreneurial business line based on women's sports; more specifically, the supply of accommodation and infrastructure for European women's soccer teams on the island of Mallorca, taking advantage of periods when there are not many tourists on the island by deseasonalizing the supply of the summer months.

In this context, the objectives set out in this work are in line with those set out in the Guide for Sustainable Tourism Agenda 2030 produced by the Spanish Network for Sustainable Development and the Responsible Tourism Institute (Azcárate et al. 2019).

In the following points, we study the references made by different authors on the subject in question and attempt to develop a methodological proposal based on the development of a specific business model (case analysis methodology), allowing us to find a solution to the stated objectives and establish conclusions in accordance with the development of the work, while also presenting its limitations.

## 2. Materials and Methods

It should be noted that the current situation of women's sports in Spain is marked by two well-known realities:

- With few exceptions, women's sports do not generate enough financial resources so that an elite athlete can dedicate herself exclusively to training and competing. It is the companies that, through different agreements and sponsorship agreements, either directly to the athletes themselves or through the different federations, support women's sports.
- In the same way, due to this lack of generation of economic resources beyond specific sponsorships at high-performance levels, basic female sports depend on the appearance of new income generators in order to reach an optimal level of development.

Given the previous situation, the theoretical framework referred to this research is defined. Its justification is based on one of the pillars indicated by the Higher Sports Council of Spain itself when referring to the development of women's sports. This pillar supports the need for companies to play a transcendental role in this development, not only through economic contributions, but also through the development of events, training plans and sustainable social actions over time (Higher Sports Council of Spain 2019).

Therefore, a concrete methodology is developed to define a conceptual framework that serves as a basis in the modeling of a case study aimed at the development of women's sports through a viable and sustainable business model: the creation of a company focused on the development of women's football on the island of Mallorca. As mentioned in the introduction, this work does not aim to assess the suitability of one tourism development model or another, but rather seeks a viable alternative for a business model that allows for sustainable, off-season and wealth-generating tourism development based on the development of women's sports.

The existing academic information on the use of the case study method in scientific research is relatively scarce, but it is usually a good methodological tool when it comes to undertaking economic impact studies (Rossi et al. 1988). By applying this methodology, it is possible to obtain the data to be analyzed from a variety of sources, both qualitative and quantitative, and it is an essential form of research in many areas of the social sciences (Nañez Alonso et al. 2021). Thus, although the case study was traditionally considered appropriate only for exploratory research, some of the works with this methodology with the greatest impact have been both descriptive and explanatory (Yin 2013). With respect to its purpose, research carried out using the case study method can be descriptive—the objective is to identify and describe the different factors that influence the phenomenon under study—and exploratory—the aim is to achieve a rapprochement between the theories included in the theoretical framework and the reality under study (Chetty 1996). In the present object of study, the research is both descriptive, in terms of the identification of sources, and exploratory, in terms of its application to the reality under study (Reier Forradellas et al. 2021).

In this specific case, the methodological approach is oriented towards the development of the procedures and elements necessary for the use of the case study method as a methodological tool for scientific research. This methodological process of scientific research includes the inductive, hypothetical and deductive phases (Martínez Carazo 2006), which allow us to reach the final objective: to study the impact of promoting an event linked to women's sports as a generator of wealth linked to tourism impact.

Similarly, there are many methodological studies that have measured the economic impact of different sporting events from different income registers, from global events such as the Olympic Games (Valiente Salinas 2014) to business models linked to tourism entrepreneurship (Blázquez 2014a); Cañada (2018).

The methodology used in this specific case is a generic model of event management on which the developed methodology can be subsequently applied. For other specific cases, it would only be necessary to adapt the reality of each event to the methodology developed (Sarabia Sánchez 1999). Three main axes provide us the necessary information to analyze the viability of the project. First, the analysis on the practice of physical exercise and sports within the current society. Second, the developed evolution of the practice of football, focusing on women's football. Third, the study of tourism as an economic sector of great importance in Spain, focusing on the different possibilities offered by sporting tourism (Butler 2001; Vrondou et al. 2018). The methodology carried out throughout this study includes the following differentiated phases:

**Phase I.** Definition and delimitation of the independent variables to study when assessing the viability of the project: conception of the general research framework.

**Phase II.** Collection of secondary sources of information on these independent variables to define a cause and effect relationship: carrying out a comparative analysis with existing data.

**Phase III.** Link the data collected with a competitive business proposal that allows the two main objectives of this study to be developed: promote women's sports through a sustainable business project.

**Phase IV.** Establish the methodology to extrapolate this data with other business models linked to the development of women's sports.

### 2.1. Analysis on the Practice of Physical Exercise and Sports within the Current Society

Physical activity and sports play a fundamental role in society, not only from a purely social point of view, but also from the perspective of public health and economic impact. With regard to health-related aspects, the public authorities themselves are making the population aware of the need to do sports for health benefits, even recommending sports as a measure to save on health expenditures based on the recommendations of the World Health Organization on the necessity and regularity of physical activity according to age groups (Claros et al. 2011). Therefore, increasing awareness about physical activity and sports has become an effective tool to promote a lifestyle that reduces the risk of pathologies that can affect people's quality of life. Not only does physical activity reduce the probability of developing pathologies, but it is also beneficial for some mental processes, school performance and the improvement of the general quality of life of people who practice sports (Ramírez et al. 2004).

The practice of sports can be considerably different depending on the country. Thus, in Nordic countries such as Sweden, Finland and Denmark, people exercise much more frequently compared to other parts of Europe (Ríos et al. 2016). They are followed, in this order, by Slovenia, the Netherlands, Belgium, Luxembourg, Germany, the United Kingdom and France. Spain ranked 17th out of the 27 countries that participated in the study. According to the study (Ríos et al. 2016), the level of practice is related to the educational and economic level of each country. Thus, we can consider the countries located in northern Europe as a reference in relation to educational and economic levels, as well as the frequency with which they perform physical exercise.

Although Spain is in a position closer to the countries that practice sports less regularly than to those at the top of the list, the practice of physical activity and sports in Spain has increased significantly in the last 40 years since studies on sporting habits have been carried out by official bodies. In 1975, barely 22% of the Spanish population aged 15 to 65 years practiced sports compared to 53% who did so in 2015 (García Ferrando and Llopis Goig 2017). In Spain, this development is due to the existence of a series of historical milestones that have favored this process (Nacimiento 2011) and have led to a change in the conception of the practice of physical activity and sports. Thus, the very establishment of a democracy and a constitution from 1976 onwards

affected the development and evolution of sports—as proof of this, we can find the following extract in the Spanish Constitution: "the public authorities shall promote health education, physical education and sport". It could be said that the State itself promoted physical education as an essential part of the development of the population, and the appropriate infrastructures were built for the practice of sports.

Alongside the legislative elements, various other factors must be added: the magnitude achieved by various sporting events, such as the Football World Cup in 1982 and the Olympic Games held in Barcelona in 1992; the important role played by sports in the media; the great visibility of sports in general and sports stars in particular, etc. All these aspects gave rise to the close relationship that currently exists between Spanish society and the practice of sports, in which Spain's elite sportsmen and women have become true media stars.

In Table 2 we can see the main reasons that Spanish people practice sports and we can appreciate that there are clear differences between those reasons depending on gender.

**Table 2.** Main reasons why people practiced sports in 2015, as a percentage of the population that practiced sports in the last year for each group.

| Main Reasons | Total | Men | Women |
|:---:|:---:|:---:|:---:|
| To be fit | 29.9% | 27.6% | 32.7% |
| Amusement or entertainment | 23.0% | 27.0% | 18.4% |
| Medical reasons | 14.8% | 12.6% | 17.4% |
| Relaxing | 13.7% | 10.9% | 17.1% |
| Like sports | 11.9% | 14.9% | 8.3% |
| As way of socializing | 2.6% | 2.4% | 2.9% |
| As personal growth | 1.8% | 1.7% | 1.9% |
| Like competing | 1.5% | 2.0% | 0.9% |
| Professional reasons | 0.7% | 0.9% | 0.5% |

Source: Our own elaboration based on data from the General Technical Secretariat of the Ministry of Education, Culture and Sport (MECD): Sports Statistics Yearbook 2020 and the Superior Council for Sports: Survey on Sporting Habits in Spain.

As seen in the table above, apart from health-related reasons—as mentioned above and recommended by the World Health Organization（2019）—from the point of view of women, the main reason to practice sports is to keep fit. For men; however, the main reason is twofold: the motivation to be fit and to have fun. Interestingly, pleasure in sports is important for men—it ranks third—but it is not an important reason for women. It could be said that women practice sports more for well-being and men for fun and entertainment.

Furthermore, as seen in Table 3, the number of women doing sports has increased (from 28.8% in 2010 to 42.1% in 2015); this increase is higher than in men (from 45.5% in 2010 to 50.4% in 2015). Although more men still practice sports than women, the increase of 13.3 percentage points in women in contrast to 4.9 percentage points in men is noteworthy.

**Table 3.** Evolution of weekly sports practice (percentage corresponding to the total population studied in each group).

| | | 2010 | 2015 |
|:---:|:---:|:---:|:---:|
| | Total | 37.0 | 46.2 |
| Gender | Men | 45.5 | 50.4 |
| | Women | 28.8 | 42.1 |

| | | | |
|---|---|---|---|
| | Between 15 and 24 years old | 57.9 | 76.1 |
| Aged | Between 25 and 54 | 40.2 | 53.2 |
| | 55 years and older | 22.2 | 26.0 |
| Level of education | Primary or secundary education | 33.2 | 39.4 |
| | Higher education or rquivalent | 54.9 | 64.1 |

Source: Own elaboration based on data from the Superior Council for Sports.

Once again, these data show that women and sports play an increasingly important and active role in society, unlike in past decades. Thus, in 1980, barely 17% of Spanish women practiced sports, while in 2015, 42% of the total already practice some sports (García Ferrando and Llopis Goig 2017). It can be seen that female sports practice has undergone an evolution in recent years, and therefore, it has become a great market opportunity for any business sector related to the practice of physical activity and sports. In addition, the fact that, in most countries, a multitude of efforts are being made in different areas—both public and private—to achieve equality between men and women, has also had an influence on sports, causing women to practice sports more and more assiduously, both from an amateur and professional point of view (Salcedo Miguel 1993). Thus, the passive presence of women in sports, as practitioners and attendees at sporting events, has become active in recent years.

### 2.2. Evolution of Football Focusing on Women's Football

The reason for linking the development of women's sports through a business model is methodological. As noted by the former Secretary of State for Sports, although the gap in sports between men and women has been reduced by 50 percent, there is still an important stretch to go, and it is a challenge of the Higher Sports Council to reverse this trend: "Women's sport takes off thanks to private momentum"(

According to the latest study on sports habits carried out on the Spanish population by the Ministry of Education, Culture and Sport, soccer is the most practiced team sport in Spain. Logically, this data is highly representative when taking into account the circumstances that characterize the practice of a team sport, such as the need for coordination and meeting of the members to practice it. In fact, due to the high number of participants required for its practice, this figure could be a handicap for the "democratization" of soccer; however, the reality demonstrates the absolute integration of soccer in all sectors of Spanish society.

Focusing on women's soccer globally, according to the Fédération Internationale de Football Association (FIFA), 30 million women play the sport worldwide (FIFA 2015). FIFA (2017) expects this figure to rise to 45 million in the next five years after the end of the Senior World Cup in France in 2019. With regard to the number of federation licenses, two distinct global areas can be defined: approximately half of the federated players (47%) are in the United States and Canada, while Europe also accounts for a high percentage of the total (44%). Only 9% of the total number of federated players are located in other parts of the world. On the other hand, only 16% of the women who play the sport worldwide are licensed and interested in participating in regulated competitions.

In Europe, and according to data provided by the Union of European Football Associations (UEFA 2017), the results obtained in relation to the evolution of women's soccer are very positive. Thus, as demonstrated by the data itself provided by UEFA, in the 2016–17 season, licenses more than doubled at the professional and semi-professional levels (3572 licenses) compared to the data recorded during the 2012–13 season (1303 licenses). An important fact is the age range in which this ratio has increased the most. Thus, in 2017, the total number of U-18 players (under 18 years of age) was 960,599, organized among 35,183 teams, in contrast to the 19,771 teams that existed in

2013. These data indicate the growth potential of women's soccer as a sport and, logically, as a possible generator of business models.

In Spain, football and indoor football constitute two modalities, whose number of licenses have increased over the last years according to the Statistics of the Federated Sport, going from 869,320 in 2012 to 942,764 in 2016. When it comes specifically to football, we can observe the increase of federated licenses according to gender over the past years in Spain, as shown in the annual directories published by the Royal Spanish Football Federation in Table 4. We can see that women's licenses have increased significantly during the past six seasons, going from 31,374 in the 2013–14 season to 47,670 in the 2018–19 season.

**Table 4.** Football and indoor football sports licenses in Spain during the last four seasons, organized by gender.

| Season | Licenses in Spain | | | | |
|---|---|---|---|---|---|
| | **Male** | **Female** | **TOTAL** | **% Male** | **% Female** |
| Season 2018–2019 | 796,614 | 47,670 | 844,284 | 94.36 | 5.64 |
| Season 2017–2018 | 781,321 | 42,543 | 823,864 | 94.83 | 5.17 |
| Season 2016–2017 | 765,818 | 40,354 | 806,172 | 94.99 | 5.01 |
| Season 2015–2016 | 720,376 | 31,831 | 752,207 | 95.77 | 4.23 |
| Season 2014–2015 | 700,526 | 29,904 | 730,430 | 95.91 | 4.09 |
| Season 2013–2014 | 682,813 | 31,374 | 714,187 | 95.61 | 4.39 |

Source: Self-made by the author based on the annual memories published by the Royal Spanish Football Federation.

However, Spain is still far away from the European countries, which, in terms of women with licenses as well as the number of teams, substantially exceed Spain's figures, as seen in Table 5. Thus, we can consider those countries as European potential nations regarding the practice women's football.

**Table 5.** Number of licenses, teams and national teams in six European countries with more female licenses in comparison with Spain.

| | Countries with More Female Licenses | | | | | | |
|---|---|---|---|---|---|---|---|
| | **England** | **France** | **Germany** | **Netherlands** | **Norway** | **Sweden** | **Spain** |
| Number of Licenses | 106,910 | 106,612 | 209,713 | 153,001 | 100,066 | 179,050 | 31,831 |
| Number of Senior Teams | 1545 | 2050 | 4456 | 2274 | 525 | 1178 | 113 |

Source: Self-made by the author based on the data published by the UEFA (2017).

In the academic field, there are studies that have considered women's sports, and more specifically the case of women's soccer, as an opportunity to generate equality in the sports field and as a possibility of seeking new alternatives for sports and economic impact (Codina and Pestana 2012; Gómez-Colell 2015;). In the same way, this possibility is equally shared by the administrations, both from the point of view of gender equality and from the point of view of sports development. In Spain, provisions in this regard have been incorporated into the legislative framework, such as Royal Decree Law 5/2015 of April 30 2015 or Law 36/2014 of December 26 2014 where the "Universo Mujer" program was declared an event of exceptional public interest (Pérez-Amor 2016). FIFA itself has developed a specific strategy for women's soccer within its objectives.

All these factors led to the analysis of the case of the proposed work to study the viability of events related to women's soccer as a potential generator of business and wealth.

*2.3. Study of Tourism in Spain: Options Provided by Sporting Tourism*

Tourism is one of the main, if not the main, strategic sector of the Spanish economy. Specifically, it contributes more than 12% of GDP and generates almost 13% of total employment in Spain. Moreover, tourism has played a very important role in the Spanish economy since its inception and has been a fundamental mechanism in the recovery from the global economic crisis that affected the country in 2008 and continued the following years (Cuadrado Roura and López Morales 2015). In 2017, Spain was the main tourist destination for non-resident European citizens, with 270 million overnight stays in tourist accommodation, 21.3% of the European Union total (Eurostat 2017). As a result, Spain became the EU country with the highest net tourism revenue with a total amount of 35.2 billion euros in 2017. Logically, this situation has also made Spain one of the countries most affected at the economic level by the COVID-19 pandemic, which has fully affected the tourism sector.

The World Tourism Organization confirmed that Spain's figures in terms of tourists have been increasing over the last few years, going from 57,464 million in 2012 (90% of them coming from the EU) to 75,315 million in 2016 (80% of them coming from the EU). In 2017 Spain received a total of 81,8 million tourists, an increase of 8.6% from the previous year. In the last year prior to COVID-19, Spain again broke its own record by reaching the number of 83.7 million tourists (1.1% more than the previous year) with an overall expenditure of 92,278 million euros (based on the data of World Tourism Organization 2019). The tourism outlook remained significantly positive for the future. Despite this, the health crisis caused by COVID-19 has led to a real collapse for the tourism sector. In the current context of uncertainty about the end of the pandemic, its impact is still difficult to estimate. The restrictions imposed on the mobility of people, together with the decline in global demand for tourism services, have had a strong impact on the sector. One only has to look at the latest data published by the National Statistics Institute (INE) to see the sharp drop in the number of foreign visitors. Specifically, foreign tourists fell by 83%, from 37 million in the summer of the previous year to 6,25 million in 2020 in the same period, which meant a drop in spending by international tourists of 86%.

The Autonomous Community of the Balearic Islands, one of Spain's most important tourist destinations, is the Spanish region most affected by the impact of the pandemic and the collapse of tourism. The number of tourists visiting the islands in the first three quarters of 2020 fell by nearly 90%. The economic impact of the crisis on the islands has been devastating, considering that during the summer period it is estimated that just over 70% of the jobs generated in the community are related to tourism. Despite this, if we pay attention to the Balearic Islands, according to the Spanish National Statistics Institute, they appear as the third destination, classified by regions that received a higher number of tourists, with 16%. Regarding these tourist's origin, most of them came from Germany, with 49.9%, followed by the United Kingdom, with 15.5%. If we analyze how foreign tourists were distributed throughout the Balearic Islands throughout the year according to the Statistical Institute of the Balearic Islands (IBESTAT 2019), we can observe that they concentrate in the months between May and September, a period considered to be high season, which reflects the touristic sector's seasonality in this area. The goals included in the Strategic Plan 2018–2020 designed by the Energy, Tourism and Digital Agenda of Spain (2018) highlight the seasonal and motivational diversification, focusing not only on sun and beach tourism, but also in the home markets' diversification.

One of the options, both to revitalize the tourism sector and to change this summer seasonality, is the development of sports tourism as a business model (Cànoves et al. 2016). The concept of sports tourism could be defined as any activity carried out by a person who travels outside their usual environment for a minimum of one night and a maximum of one year with the main objective of performing a sporting activity or attending a sporting event (González Requena 2015). However, the convergence

between tourism and sports is not without controversy, and two different concepts have emerged: sports on vacation or sports vacations. In the case of the present study, reference will be made to this second concept. Another definition given by Standeven and De Knop relates sports tourism to all active or passive forms of sporting activity in which one participates in a casual or organized manner, whether for commercial, business or other reasons, but which necessarily involves travel away from one's usual place of residence or work (Latiesa and Paniza 2006). Other authors refer to individuals or groups that actively or passively participate in competitive or recreational sports during a trip away from their usual residence (Gammon and Robinson 1997).

On the other hand, sports tourism can also be classified according to the type of public it attracts, on the basis of three characteristics: those that generate a public that is limited to attending or following a live sports activity, those that generate a type of public that includes active tourism related to sports facilities (visits to stadiums, museums, representative places in the field of sport, etc.) and those that involve the active practice of a specific sporting activity (Díaz and García 2015). In Spain, according to the Sporting Habits in Spain Survey conducted by the Ministry of Education, Culture and Sports of Spain, the entry of international tourists increased in 2019 in comparison to previous years because of reasons related to sports. However, it does not represent a significantly high percentage in comparison with the total number of entries, less than 2%. We can observe some information about the tourist's expenses in Table 6.

**Table 6.** Entries and total expenses of international tourists due to reasons related to sport.

|  | 2015 | 2016 | 2017 | 2018 | 2019 |
|---|---|---|---|---|---|
| **Entries (thousands)** | **965.1** | **1413.5** | **1272.8** | **1473.1** | **1512.5** |
| Percentage in relation to the total entries of international tourists due to leisure, entertainment or vacation | 1.7 | 2.2 | 1.8 | 2 | 2.1 |
| Percentage in relation to the total entries of international tourists | 1.4 | 1.9 | 1.6 | 1.7 | 1.8 |
| **Expenses (millions of euros)** | **877** | **1393.8** | **1,255** | **1400** | **1434** |
| Percentage in relation to the total expenses of international tourists due to leisure, entertainment or vacation | 1.5 | 2.1 | 1.7 | 1.8 | 1.8 |
| Percentage in relation to expenses of international tourists | 1.3 | 1.8 | 1.4 | 1.6 | 1.6 |

Source: Own elaboration based on data from MECD. Sporting Habits in Spain Survey 2017, 2018 and 2019 and Sport Statistics Yearbook 2020.

According to the above data, there is room for a change in the strategy aimed at finding market niches within the tourism sector. At this point it would be a new tourism model that would benefit the idea of sports and physical activity as a strategy for Spanish tourism (Díaz and García 2015). This new perspective requires the collaboration of both the public sector and the private sector, becoming a space where both must interact. In 2017, Spain already attracted more than 10 million domestic and international tourists related in some way to sports, which generated revenues close to 14 billion euros ,according to the tourism expenditure survey conducted by the Institute of Tourism Studies (EGATUR 2019. In addition, following the Spanish model as a complementary tourism offer, the sports sector has established itself as an important

complementary attraction for areas with a traditional tourism offer. However, this offer must undoubtedly meet a whole series of requirements to become a real line of business: it must be an integrated, attractive, playful, well-promoted, profitable and sustainable activity.

It should be noted that the above definition of sports tourism includes:

- Travel to a place other than the usual place of residence or work;
- The purpose of the trip is mainly recreational (leisure), but related in some way to sport, either in artificial facilities or in the natural environment;
- Physical activity may be performed, or sporting events may be observed (passive-active);
- There may or may not be competitive purposes.

This concept makes it possible to clearly delimit the profile and type of client (Kurtzman and Zauhar 1997). In fact, this approach is the basis of the methodology applied in this study, favoring the development and impact of women's sports as a business creation base linked to the concept of sports tourism in areas with a wide traditional tourist offer (Medina and Sánchez 2005). As the author García Ferrando points out "sports habits are increasingly linked to lifestyles, creating a complex matrix of leisure and sports behaviors, which demand a no less complex offer of services that give users satisfaction" (García Ferrando and Llopis Goig 2011).

As shown in Table 7, in the months of January and February, on the one hand, and the months of October, November and December, on the other hand, many hotels in Mallorca do not open their doors because of the seasonality of tourism and the low season. This is reflected in the percentage of available hotel beds, which hardly changed in the period analyzed (2012–2018).

**Table 7.** Percentage of available hotel beds and occupancy rate of available hotel beds in the Balearic Islands broken down by months (2012–2018).

| | AVAILABLE HOTEL BEDS | | | | | | | OCCUPANCY RATES OF AVAILABLE HOTEL BEDS | | | | | | |
|---|---|---|---|---|---|---|---|---|---|---|---|---|---|---|
| **Months** | **2012** | **2013** | **2014** | **2015** | **2016** | **2017** | **2018** | **2012** | **2013** | **2014** | **2015** | **2016** | **2017** | **2018** |
| January | 8.3 | 7 | 5.7 | 5 | 5.3 | 5.4 | 5.5 | 32.3 | 38.9 | 29.8 | 35.6 | 38.6 | 37.2 | 35.9 |
| February | 15.5 | 12.9 | 11.3 | 10.7 | 10.5 | 11.2 | 11.8 | 43.8 | 45.9 | 42.7 | 46.6 | 50 | 46.1 | 48.4 |
| March | 22.9 | 25.4 | 19.8 | 19.2 | 22.5 | 19.5 | 20.3 | 54.4 | 53.7 | 51.1 | 52.4 | 58.2 | 55.6 | 55.5 |
| April | 38.2 | 37.3 | 37.4 | 38.6 | 38 | 37.7 | 39.9 | 58.3 | 58 | 62.4 | 64.1 | 65.5 | 70.8 | 63.1 |
| May | 88.2 | 88 | 89.1 | 89.7 | 90.1 | 92.9 | 93.5 | 60.6 | 64.5 | 59.3 | 61.6 | 69.9 | 69.6 | 68 |
| June | 95.5 | 96.3 | 97 | 95.6 | 96.8 | 96.9 | 97.7 | 81 | 80.5 | 80.2 | 80.2 | 84.6 | 84.8 | 82.9 |
| July | 97.3 | 97.6 | 97.2 | 96.5 | 96.8 | 97.2 | 98.1 | 90.1 | 89.1 | 87.4 | 89 | 91.5 | 90 | 89.5 |
| August | 97.7 | 97.7 | 97.6 | 97 | 97.4 | 97.3 | 97.8 | 92 | 91.5 | 91.3 | 93.1 | 92.9 | 90.5 | 90.3 |
| September | 97.3 | 97.1 | 97.1 | 96.2 | 97.2 | 97.2 | 97.8 | 82.6 | 82 | 82.1 | 83.4 | 86.6 | 84.6 | 83 |
| October | 73.4 | 73.7 | 74.6 | 78.9 | 79.3 | 80.5 | 80.4 | 58.9 | 60.9 | 59.7 | 61.1 | 67.1 | 66.5 | 64.8 |
| November | 7.9 | 7.6 | 6.8 | 6.5 | 7.9 | 8.2 | 7.8 | 43.8 | 41.1 | 43.6 | 46.3 | 50.4 | 50.3 | 44 |
| December | 5.8 | 5.7 | 4.1 | 4.8 | 5.4 | 5.4 | 6.2 | 38.5 | 31.7 | 39.8 | 38.4 | 39.5 | 39.2 | 41.7 |
| **Total** | 54 | 54 | 53.2 | 53.3 | 54 | 54.2 | 54.7 | 74.4 | 74.9 | 74.2 | 75.7 | 79.2 | 78.4 | 76.8 |

Source: Own elaboraSource: Own elaboration based on data from Spanish Ministry of Tourism 2019

This implies that the occupancy rates of available hotel beds during those months of the low season are very low. The attraction of women's teams from northern Europe during these months could help raise both the percentage of available hotel beds and the occupancy rates of available hotel beds, having a very positive impact on the Balearic Islands.

Table 8 shows how the target markets of the proposal to develop women's sports as a source of business linked to tourism already have a great economic impact on the

Balearic Islands. Thus, Germany, the United Kingdom, the Nordic countries and the Benelux have a large tourist presence in the Balearic Islands (Mallorca being the main reference) and a very important impact in terms of wealth. In terms of average spending per day and per tourist, Nordic and Benelux visitors make an average spending per day and per tourist that exceeds 160 euros (165.2 and 169.2 euros, respectively).

**Table 8.** Expenditure of tourists according to country of residence (2018).

| Countries | TOTAL EXPENDITURE (MILLIONS EUROS) | | DAILY EXPENDITURE PER PERSON (EUROS) | |
|---|---|---|---|---|
| | 2018 | % Var. 18/17 | 2018 | % Var. 18/17 |
| Germany | 4,764.3 | −3.9% | 144.1 | 9.70% |
| UK | 3,379.4 | 3.00% | 148.4 | 7.50% |
| Nordic | 1,165.1 | −2.30% | 165.2 | 2% |
| Benelux | 1088.6 | 13.20% | 169.2 | 6.20% |
| Italy | 715.4 | 6.60% | 141.8 | 11.40% |
| France | 707.9 | 24.40% | 143.4 | 9.60% |
| Switzerland | 583.4 | 10.20% | 153.5 | 3.70% |
| Others | 2,117 | −1.50% | 212 | 9.40% |
| Total International | 14,281.1 | 1.50% | 155.8 | 8.10% |
| Domestic | 1,544.2 | 7.40% | 90.9 | 1% |
| **Total** | 16,365.3 | 2% | 146 | 6.70% |

eSource: Own elaboration based on data from Spanish Ministry of Tourism 2019.

As can be observed, the German market is the one with the largest number of tourists. Focusing on this data, in 60% of the cases, the German tourist has the intention to practice sports during his vacations, for 10% of cases, sport is the fundamental motive for the realization of his trips, and for 2%, the only reason to travel is to attend a sporting event as a spectator or as a participant (Freyer and Grob 2002). As a conclusion to this part, at the last World Conference on Sports and Tourism, the following points were highlighted (OMT-WTO 2001):

- Sports have become an important factor in the offer of tourist destinations, allowing them to differentiate themselves and be more competitive in the international framework;
- Professional athletes have become an increasingly important market for tourist destinations;
- Major sporting events are an excellent image campaign for a destination that wants to promote its tourism aspect;
- Young people invent and discover new forms of tourism and new sports, which often become popular and can become mass phenomena.

### 3. Literature Review

As already mentioned in this paper, the interconnection between tourism and sports is evident in advanced societies. This interest is reflected in the efforts made by both the public and private sectors to strengthen this connection (Latiesa and Paniza 2006). In the academic field, scientific production referring to sports tourism is very recent, as the official journal *Sports Tourism International* was first published in 1993. However, its evolution and development linked to the economy has led to the search for new business models. Tourism cooperation between different actors, both public and private, has become especially important as one of the keys to success. Tourism development cannot be a process characterized by the inertia brought by the arrival of tourists to the destination; rather, it must be the product of a reflexive process on the



part of the destination territory and the actors involved in shaping the tourism system of the territory (Bramwell and Lane 2000).

The role of the local stakeholders that participate directly or indirectly in the tourism product of each territory must necessarily be active, including the private companies involved (Valls 2003; WTO 2001). These contributions should be aimed at achieving sustainable development of the territory. One of the great challenges of new tourism relations, especially in places as overcrowded as the Balearic Islands in general, and Mallorca in particular, is to implement responsible and sustainable tourism models in the long term (Rivera Mateos 2011; Butler 2011). In the same way, through the proposal identified in this study, measures can be offered referring to the deseasonalization of tourism, which is a fundamental aspect when it comes to maintaining adequate sustainability in tourism (Castel Gayán and Lacasa Vidal 2012).

Work based on innovation in the tourism sector has already been studied in the academic field. These studies have been carried out from both a theoretical and a practical perspective. As pointed out in the Introduction Section, it seems clear that tourism continues to be the fundamental pillar and mainstay of the economy of traditional destinations such as Mallorca, but it urgently needs to modernize its growth models (Santoyo Calderón 2015). Within this modernization, experts include the so-called "Barcelona 92 model", where numerous destinations use sports as a way of activating and improving their tourism management (Menchen 2019). Logically, for this synergy between tourism and sports to take place, it is necessary to have a sufficiently strong base in terms of infrastructure, facilities and traditional tourist management (all these points exist in the case of Mallorca; Peláez 2019).

In the specific case of Spain, with the model of sun and beach tourism showing signs of exhaustion, it is necessary to look for alternatives in order to continue to occupy a priority position as a world tourist destination. In this sense, sports tourism can be a key part of these alternatives (López 2019). The current Secretary General of the World Tourism Organization, Zurab Pololikashvili, pointed out that diversification of the tourism sector is welcome as it spreads the economic and social benefits of tourism more widely, and niche segments, such as sports tourism, are the ones that are really gaining strength and are likely to grow in size and importance in coming years (López 2019).

Even the most recent studies relating the situation of traditional tourist destinations to the COVID-19 pandemic (specifically for the Balearic Islands) point to tourism innovation models with private sector management capacity as essential for the recovery of tourism activity (García Ramis 2020).

In this sense, as already mentioned, in 2019, more than 10 million foreign tourists were expected to arrive in Spain for sporting reasons, generating revenues of more than 12,000 million euros (Palomar et al. 2020). This effect is not just seen in Spain; according to the World Travel and Tourism Council, this sports tourism trend already accounts for 25% of tourism revenues.

Finally, it is important to summarize the overall objectives that should be included in a new model of sustainable tourism (Santoyo Calderón 2015):

- Improve seasonality management;
- Improve the diversification of markets and products;
- Improve awareness of the destination and its products, and thus, its brand image;
- Increase the number of visits to the destination, especially overnight stays;
- Increase the average length of stay;
- Increase tourist spending at the destination;
- Increase tourist satisfaction during their visit;
- Increase loyalty by increasing the rate of repeat visits.

Finally, since this is an entrepreneurship project, it is interesting to highlight , if only as an example, similar models that are working in Spanish localities (only a couple of cases will be listed as a sample):

- The company Football Impact (www.footballimpact.com, accesed on 18 March 2021) is one of the leaders in the sector (mainly focused on men's football) and operates on the Spanish "Costa del Sol" (city of Malaga) with a turnover rate growing at around 20% per year (with an average turnover of 14 million euros in 2019);
- The "Costa del Sol" itself, through the "Sport Destination" program run by the public organization "Turismo Costa del Sol", is developing a business model that foresees average stays of more than 227 teams in 2019;
- As an example, in the city of Seville, the company Spain Soccer Academy (www.spain-socceracademy.com accesed on 18 March 2021) operates a business model based on the stay of amateur players and teams for a specific period of time. Futbollab (www.futbollab.com accesed on 18 March 2021) combines the stay of teams with coaches, operating mainly in the city of Barcelona. Another example is the company Fútbol DF (www.futbol-df.com accesed on 18 March 2021), which also provides stays for football teams (in this case focusing mainly on youth teams), although it does not operate in a fixed location.

On the basis of these approaches and this theoretical framework, the following section develops an analysis of the expected results. In this type of analysis, the main problem always comes from the revenue side, as it is necessarily based on forecasts. Expenditure is easier to estimate as it is based on direct items that can be known, with a certain margin of error. The methodology used here is the classic business model methodology, with a forecast of the revenues and expenses expected to be generated by the business model and a forecast of results. Dependent variables are estimated in the revenue items based on two key concepts:

- Number of teams that will demand the services;
- Average revenue to be generated by each team.

From these two dependent variables, a series of cost estimates are generated on the basis of our own forecasts of the number of participating teams.

## 4. Results

On the basis of all the points analyzed so far, we believe that there are sufficient reasons to try to respond to the objectives set out in this study: to develop an opportunity to generate business and wealth that integrates the promotion of women's soccer as a measure of integration and equality linked to sustainable tourism development. Before looking for the direct impact that the implementation of a specific company would have, the present study should serve as a basis for measuring the following associated objectives, based on the three items studied in the previous section:

- Examine the evolution of the female population in terms of sports practice and, specifically, in relation to the sport (football) and location (Mallorca) proposed in the present study;
- Analyze if the measures taken to promote women's sports can have the desired effects through the comparison of the results obtained on the basis of a work methodology;
- Investigate lines or proposals of action towards which future measures should be promoted to continue promoting female sports.

The "Universo-Mujer program" developed by the Spanish government (Bramwell and Lane 2000) consists of several projects that seek the dissemination and promotion of women's sports through a series of pillars where companies play a transcendental role, not only through financial contributions, but also through the development of

events, training plans and social actions. The previous reasons demonstrate the scientific interest of the case study proposed in this study.

The proposed business model meets the national demand and need of clubs (professionals and amateurs) and women's soccer teams that want to seek new accommodation and training options outside of their usual places (usually their own areas of location). We propose stays for short periods of time to serve as preparatory stages prior to a particular competition, taking advantage of the climate of Mallorca, ideal for this activity during many months of the year due to the optimal weather conditions, especially in relation to the countries of origin.This situation can allow the respective teams to improve their preparation in order to achieve maximum competitive efficiency before or after a specific event. It should be noted that this activity is recurrent, so that teams using it can return every season, with the consequent impact on the sustainability of the proposed business model. Logically, this possibility would also be open to non-professional clubs that wish to carry out joint stays in places such as Mallorca, and in these cases, the opportunity would not be so focused solely on the sporting approach, but also on the tourist experience.

The proposed service includes all those aspects that must be implicit in the proposed business model, which, as a whole, will increase profitability. In this way, the travel, accommodation and sports facilities to be used, the possibility of hiring rival teams and any other sports services that users require are be taken into account. Similarly, the possibility of contracting other non-sports entertainment, culture and leisure services is included. This is why it was indicated that the development of this type of tourism can have a direct economic impact on all economic areas of Mallorca.

As already indicated, from the two dependent variables based on the forecast of participating teams and the forecast of revenues per team, a series of cost estimates was generated. In order estimate all these data in the most realistic way, we took into account the realities of the companies that are already operating in the sector (many of them mentioned above), adapting their reality to our model.

Regarding the sports services offered, the main reference is the availability of professional football fields, including both natural and artificial grass. Most of these facilities are located in different municipalities in the area known as "Part Forana", which also coincides with the most touristic area of the island. Because of the size of the project, agreements need to be reached with the different municipal terms for the rental of the different necessary facilities. In the same way, the available offer can be enhanced with the use of and rental agreements with private facilities. In addition to the football fields themselves, other sports facilities, such as swimming pools, sports centers, spas, golf courses, etc., should be made available to the clubs. It must be taken into account that a large majority of hotels, due to the fact that Mallorca is one of the reference destinations in Europe as a whole, already have these types of facilities, so the contracting facilities make the project highly feasible. In the same way, it is possible to organize training sessions, friendly matches and tournaments with other local or international women's teams that are also hosted on the island at the same time. More specifically, the services for organizing training sessions and tournaments include the following aspects:

- Hydration fluids for the teams;
- Equipment and local sporting materials located in the facilities;
- Referees who are members of the Balearic Football Federation;
- Opposing teams appropriate to the characteristics established by the technical staff.

Regarding non-sporting services, accommodation in the hotels of the island that offer all the facilities is ideal, so that, while the teams are staying in Mallorca, they can concentrate on the sporting preparation. There are four- and five-star hotels belonging

to the hotel group "Iberostar Hotels & Resorts", which are located throughout the whole island's coast. The following services should be included:

- Accommodation type (all-inclusive, full board, half board) with menus designed by sporting nutritionists;
- Availability of hotel's facilities (meeting rooms, material-storage rooms, spa, gym, etc.);
- Medical services.

Displacements at the arrival and departure from the island, as well as any other displacements made during their stay are included in the proposed services. The offer also includes other services upon teams' request during their concentrations. Those services include leisure activities (trekking, horse riding, mountain biking, climbing walls, etc.), cultural activities (touristic bus tours with a guide; Segway tours, bicycle and electric scooter tours; shopping tours and shopping centers) and recreational activities (aquarium, safari, zoo, shows and events, paintball, bubble-football, etc.) to enjoy their rest and free time. The agents involved in the development of those activities are the following:

- Balearic Football Federation: can provide the referees needed for friendly matches and to look for opposing teams;
- Local sporting staff from localities in the "Part Forama" can oversee booking local sporting facilities and equipment;
- Company of computer services for web page and content management;
- Hotel chain Iberostar Hotels & Resorts for accommodation services;
- Bus company Transunion Mallorca for displacements;
- Services company for other leisure, cultural and recreative activities;
- Company staff, whose availability is currently being analyzed;
- Tax and work consultancy-assessors for accounting, tax and work management.

From this analysis, we can see a list of services and agents participating in the start-up of the business related to women's football and its implementation in Mallorca. This analysis serves as a basis to replicate the proposed activity in other parts of the island of Mallorca, taking into account the premises that we have indicated.

### 4.1. Client Profile

As we already pointed out in this paper, there are numerous academic contributions referring to case studies applied to the field of tourism, in which economic development linked to sustainable tourism was considered. Many of them referred to specific geographical areas, as in our case, as we refer to the city of Mallorca (Granovetter 2005; Hiwasaki 2006; Murphy and Murphy 2004), while others have referred to projects in which private initiative has been fundamental (Tosun 2005; Novelli et al. 2006) and others have directly linked tourism and sports (Blázquez 2014b); Murphy et al. (2000); Derry et al. (2004).

The recipients of the service offered are, as already mentioned, women's football teams, which belong to clubs (regardless of the category to which their leagues belong), national teams or lower categories. Regarding the type of client profile, teams that come from countries in which women's football has a greater implementation and that suffer from adverse weather in the months outside the summer period are the primary clients sought for this proposal. In this way, it is important to choose countries that have a greater number of federative licenses and that, at the same time, have the largest number of clubs and teams in Europe. Considering this analysis, the main applicants for the service should come mainly from countries such as Germany, England, Holland, Sweden, Norway and France. It is very important to point out that the countries with the highest number of federation licenses are also countries that traditionally bring a large number of tourists to Mallorca, making it a recognized and highly regarded destination. However, other nationalities should not be neglected and, in the

future, we hope to spread the proposed model to other continents such as America or Asia.

The average stay's duration was 9 days, taking as a reference the data corresponding to men's football teams that visited the Costa del Sol in 2016. Regarding temporality, the proposed service is available all year long. Nevertheless, the proposal tries to establish most of the visits in three periods:

- November, December and January: taking advantage of the periods free of national-league competitions regarding the main European leagues;
- February and March: months prior to the beginning of the league competitions in Norway and Sweden.
- May and June: the final phases of the European competitions and world cups for national teams (senior and inferior categories) take place in May (European Championship U17), June (world cups) and July (European Championship U19). The preceding weeks are usually used by the national coaches to prepare for those competitions.

To develop the customer profile, a similar activity that has already taken place in another territory of Spain: the Costa Blanca, was analyzed. In this place, the average stay per team was 9 days. As it is a logical term, these data were taken as the basis of this study. The method used to select the months and periods was from the analysis of women's football leagues in European countries, especially the rest periods. This period and form of analysis could be perfectly repeated if this same activity were carried out in another part of the island.

### 4.2. Setting Price of Service

The budget study is carried out by defining a clear cost and income model for the different possibilities of services chosen by each club. Logically, a clear methodology should be carried out to link this study to the different models of clubs and services that could contract the services. Thus, on the expense side, it is necessary to add the total of the standard costs of the services contracted for each category of equipment. In this way it is possible to calculate the corresponding percentage of profit on the total costs (depending on the category of each club).

Those profit percentages are established according to the following aspects.

- Team or club position:
  - High profit (between 18% and 22%): a senior club that is taking part in the highest national category of their country or a senior national team;
  - Medium profit (between 15% and 18%): a senior club that is not taking part in the highest national category of their country or a national team of the inferior categories U23, U19 and U17;
  - Low profit (between 12% and 15%): an amateur club or national team of the inferior categories U16 and U15.
- Number of people travelling:
  - More than 35 people, the profit percentage will be reduced by 1%;
  - Between 25 and 35 people, the profit percentage will be reduced by 2%;
  - Less than 25 people, the profit percentage will be reduced by 3%.
- Promotions and discounts to clubs that present one of the following characteristics:
  - Usual user: having booked the service previously;
  - Recommendation to future clients: teams recommended by a client;
  - Same club: if several teams from the same club book the service.

It should be noted that the estimation of these data was based on the different realities analyzed in the sector due to the different level of professionalization of the participating teams.

### 4.3. Communication and Sponsors

The clients are European, so it is necessary to design an effective communication strategy to promote the proposed business model. This strategy should focus on the following communication aspects.

- Offline publicity:
    - o Radio commercials in local radio stations in different languages that are aimed at tourists, including Das Insel Radio in German and Mallorca Sunshines Radio, or Radio One Mallorca, in English;
    - o Written press including newspapers aimed at British and Germans clients, such as Majorca Daily Bulletin and Mallorca Zeitung (written and online versions).
- Online publicity:
    - o Web pages;
    - o Social networks including Facebook, Twitter, Linkedin and Instagram;
    - o SEO media positioning campaigns.
- Sales promotion: Participation in different touristic fairs to give visibility to our products, such as "International Torism Berlin" (ITB) in Berlin, "Feria Internacional de Turismo en Madrid" (FITUR) in Madrid, World Travel Market in London, etc. In the same way, attending specific fairs dedicated to the sports and tourism sector that are held in the countries should help to identify the main group of clients.
- Personal sale: Promotion and dissemination activities should be organized for different European tour operators so that they can offer this service in their countries of origin. In this way, the aim is to incorporate the proposed business model as a speciality of different tour operators.

The communication strategy is clearly and precisely developed. In this way, this list of activities is presented as a list to follow and can be repeated by other entrepreneurs.

The role played by sponsors in relation to women's football is becoming more and more important. This is happening in relation to men's football. In this sense, the project seeks to incorporate the sponsorship of external companies that see this service as a commercial opportunity. Financial support for some of the activities carried out that are a source of sponsorship would be desirable. In this way, it can be possible to carry out a sustainable and long-term sponsorship plan with agents involved in the same sector and business model (mainly hotels and other companies in the tourism field) to reduce the costs of the services offered, decrease the need for financing and increase the profit margin. In our proposed model, all these potential sponsors will be offered personalized rewards that take into account the visibility of their brand within the marketing action to be developed. It must be taken into account that with this dissemination, as they are also businesses linked to the tourism sector, they can see their direct impact increase on an audience that is also part of their target group. After this process, they should seek the presence of sponsors, both national and foreign, linked to the clubs' countries of origin. In particular, those with interests in countries such as Germany, England or the Nordic countries should be targeted.

### 4.4. Financial Study

Next, the basic financial and economic information about the project is offered as a basic element to determine its viability.

Investment expenses include the initial outlay required to establish the company, which includes the acquisition of the minimum resources for the subsequent activity's development, as seen in Table 9.

**Table 9.** Estimated annual cost.

| Concept | Cost |
|---|---|
| Formation expeses | 400.00€ |
| Office IT | 750.00€ |
| Web page | 450.00€ |
| Several materials | - |
| Publicity | 4,000.00€ |
| Total Cost | 5,600.00€ |

Source: Own elaboration.

Fixed expenses include the costs agreed with others, which have to be paid by the company on a monthly or annual basis, so as to ensure an optimal service performance, as seen in Table 10.

**Table 10.** Estimated annual fixed expenses.

| Concept | Cost |
|---|---|
| Insurance Policies | 700.00€ |
| Telephone and Internet | 900.00€ |
| Company Vehicle | 2,500.00€ |
| Office Renting | 7,200.00€ |
| Own personal wage | 24,400.00€ |
| Web person | 1,200.00€ |
| Total Cost | 36,900.00€ |

Source: Own elaboration.

Operating expenses include costs that arise as a result of developing a certain service hired by a club. Those have to be paid by the others, who should be hired to provide the services agreed upon with the club, as seen in Table 11.

**Table 11.** Operating expenses estimated by model service.

| Concept | Cost |
|---|---|
| SPORTING SERVICES | |
| Renting sporting local and private facilities | 1,300.00€ |
| Coordination and resources for training sessions | 70.00€ |
| Coordination and resources for friendly matches | 450.00€ |
| NON-SPORTING SERVICES | |
| Transfers and displacements | 3,500.00€ |
| Accommodation (includes food and other hotel services) | 16,200.00€ |
| OTHER SERVICES | |
| Guided tours | 600.00€ |
| Visit to museums and popular places | 900.00€ |
| Leisure activities (paintball/bubble-football) | 2,500.00€ |
| MALLORCASOCCER COMPANY SERVICES | |
| External staff compensation | 1,000.00€ |
| **TOTAL COST** | **26,520.00€** |

Source: Own elaboration.

Sponsor Income: according to the proposed sponsorship plan, it is estimated that it would be possible to obtain a reduction in operating expenses from the agents involved as compensation for the advertising promotion carried out. A reduction of 1% on total expenses can be considered. Thus, Table 12 shows the estimate of annual income, which comes from agents involved in the product and from external companies. As noted above, these cost items were based on the actual stay forecasts.

**Table 12.** Total annual income estimate from sponsors.

| Agents Involved | | External Companies | | Total Sponsorship |
|---|---|---|---|---|
| Annual operating expenses with involved agents | 583,440.00€ | Economic contribution | 5,000.00€ | 10,834.40€ |
| Contribution regarding sponsorship (1% from expenses) | 5,834.40€ | | | |

Source: Own elaboration.

Clients act as the main source of income for the company. Clients who agree on the service must pay 50% of the budget in advance, which will help the company advance the payment of the externally hired services, although they will try to agree on a payment subsequent to the service. As previously described, in the proposed plan, the company applies a different profit percentage (between 12% and 22%) according to the main criterium of the club's positioning (depending on whether the team is a professional, amateur or a national team, and to which category it belongs). Thus, they can establish an average profit percentage of a 17%, so as to calculate the profit obtained from a model service. In Table 13, the estimated profit that the company can obtain if we apply the average percentage of a 17% will be 4508.40€ per service.

**Table 13.** Estimated income obtained from each service received and profit obtained on each service.

| | |
|---|---|
| Total cost of service provided | 26,520.00€ |
| Commercial profit of 17% on the total cost | 4,508.00€ |
| Income received by a club/client | 31,028.40€ |

Source: Own elaboration.

We provide an estimate of services, expenses and income. As stated before, the main period during which the services develop is November and June. Consequently, we estimated the number of services provided during the first year. Table 14 shows the foreseen periodization of services together with the expenses and income estimate for the company.

**Table 14.** Monthly estimate of services, with the operating expenses of those services and the income they generate.

| Month | Estimated Monthly Services | Operating Costs | Incomes |
|---|---|---|---|
| November | 3 | 79,560.00€ | 93,085.20€ |
| December | 3 | 79,560.00€ | 93,085.20€ |
| January | 3 | 79,560.00€ | 93,085.20€ |
| February | 3 | 79,560.00€ | 93,085.20€ |
| March | 2 | 53,040.00€ | 62,056.80€ |
| April | 2 | 53,040.00€ | 62,056.80€ |
| May | 2 | 53,040.00€ | 62,056.80€ |

| | | | |
|---|---|---|---|
| June | 1 | 26,520.00€ | 31,028.40€ |
| Rest of year | 3 | 79,560.00€ | 93,085.20€ |
| Total | 22 | 583,440.00€ | 682,624.80€ |

Source: Own elaboration.

To these annual operating expenses must be added the annual fixed expenses of the company, which are reflected in Table 15. In this way, the total expenses that the company must pay during the first year of its constitution can be obtained. Investment expenses that can be financed with personal funds without involving an excessive outlay are excluded from these total expenses (that is, it is considered that the partners will always have the capacity to meet these expenses).

**Table 15.** Annual estimated profit depending on the annual estimate of expenses and incomes.

| | |
|---|---|
| Operating expenses depending on annual estimate of services | 583,440.00€ |
| Estimated annual fixed expenses | 36,900.00€ |
| Income received by clients/clubs depending on annual estimate of services | 682,624.00€ |
| Income-Expenses | 62,284,80€ |

Source: Own elaboration

As can be observed, after the first operational year, if the estimations regarding the number of services are met, the income obtained from clients other than the entity's costs will provide a profit of 62,284.80€. We estimate that, during the second and third operational years, the amount of annual services will increase due to the company's popularity in the Nordic Countries and to the influence on the home markets generated by the teams that visit in the first operational year. Certain defined actions regarding the marketing mix (services, prices, communication and distribution) will help this profit increase. Furthermore, we believe that an increase in the income generated from the company's sponsors will also take place. This growth forecast can be seen in Table 16.

**Table 16.** Estimate of services, income and expenses in the following year.

| | Year n | Year n+1 | Year n+2 |
|---|---|---|---|
| Number of services | 22 | 25 | 28 |
| Clients income | 682,624.80€ | 775,710.00€ | 868,795.20€ |
| Sponsors income | 10,834.40€ | 12,630.00€ | 14,425.60€ |
| Total Expenses | 620,340.00€ | 699,900.00€ | 779,460.00€ |
| **Result** | **73,119.20€** | **88,440.00€** | **103,760.80€** |

Source: Own elaboration.

It can be seen that the positive result increases year after year thanks to the clients' increase and the increasing income generated by them. These positive results can be used to invest in the company in order to constantly improve the service and attention provided to our clients. In view of the above results, we demonstrate how this business idea could generate a benefit to the private company that decides to develop it; moreover, if several actions such as the one described in this article are carried out by several businessmen, they could help to counteract the effect of seasonality on tourism in the Balearic Islands, which causes low tourism in the months of low season. All this thanks to the attraction of women's football to the Balearic Islands.

This business model can be replicated by different companies and organizations, which would make it possible to achieve the objectives set out at the beginning of the study. In short, the proposed business model follows a strategy focused on the niche

market of women's soccer, especially in Europe, which can contribute to ending the tourist seasonality of the island. The target market focuses on teams from Germany, England, Holland, Sweden and Norway, which, in addition to being teams with European potential in terms of the number of female licenses, are the countries that bring the largest number of tourists to Mallorca.

The sporting and extra-sporting services that developed in the proposed model are aimed at amateur women's soccer teams, professionals and even national teams that wish to spend a short stay in Mallorca coinciding with periods in which the European weather makes it difficult to play soccer in their countries of origin. This model means that a far greater volume of demand can occur during the traditionally medium and low tourist seasons, which will favor the deseasonalization of the tourist component.

It is important to note, as a conclusion to this point, that the limitations of the analysis of the results are themselves limitations that will also be raised in the conclusions section. As in any business model, despite being based on a clear theoretical framework and the realities of other entrepreneurial projects, all performance forecasts have a risk factor that does not depend solely on the promoters themselves. For example, the effect of the COVID-19 pandemic would have a very clear negative impact if this model was considered before 2020. These limitations are present in any academic work based on business forecasts. On the other hand, many of the data used in the reference framework are pre-COVID-19 data, since the reality of a sector such as tourism would be completely distorted if data from 2020 and 2021 were used (as already noted, overall tourism suffered an overall decline of more than 80% in Mallorca).

## 5. Conclusions

As indicated throughout the development of the paper, it seems that established traditional tourist destinations could be losing importance in favor of new emerging destinations (which are often less overcrowded and with better price offers). The so-called "tourism value chain" has also changed. The tourist has become the main actor in the whole process, demanding a high degree of quality and personalization in the contracting of tourist services. In the same way, a type of tourism that combats seasonality (which is essential, as it is literally impossible to increase the number of tourists in the high season in places like Mallorca) is necessary, which promotes a sustainable model focused on a premium tourist profile that, in addition to bringing economic benefits, also brings prestige and media coverage to the destination. All of these issues are dealt with and developed in the work, presenting an entrepreneurial model that takes up the needs raised by the literature consulted, and which includes an integrated plan for the development of women's sports.

This study initially presented a series of realities from different social spheres:

- Despite the growing incorporation of women into the world of sports, both at amateur and professional levels, there are still evident and significant differences with regard to aspects such as equality, integration and social repercussion. As the study has shown, one of the best ways to achieve the objectives of equality is to give more vision and more impact to women's sports in all possible areas;
- The Spanish tourist reality in general, and that of the Balearic Islands in particular, is marked by overcrowded sun and beach tourism, which is unsustainable from an environmental point of view, and highly seasonal in the summer months. One of the objectives that was set by all the institutions involved is to bet on new forms of tourism that are innovative, sustainable and deseasonalized. This point is even more necessary due to the implications of the COVID-19 pandemic on the tourism sector, where the impact has been brutal;
- Public-private integration as a generator of business opportunities in the definition of a new tourism model has also become one of the priorities of the post-

COVID-19 tourism sector. The economic impact of a business model based on sustainable and deseasonalized tourism is essential in an environment such as Mallorca.

All of the above aspects are addressed in this work, combining the generation of wealth from the private sector with repercussions on the public sector through job increases, indirect income, increased revenue, improved international tourist perception, increased hotel occupancy, etc. It is a model that can be extrapolated to any type of organization and format, so that the methodology applied can be perfectly replicable and exponentially increase the achieved objectives.

The proposed contributions should not only be assessed from the developer's own point of view, but also from a broader point of view:

- For its contribution to local social development with the arrival of tourists from different parts of the world, giving dynamism and modernity to the local community;
- For its contribution to the generation of wealth and employment;
- For its knock-on effect on other sectors, making the sector a global economic force.

The increase in the practice of sports among populations is a global reality. In the case of Spain, the increase in the practice of sports has grown exponentially in the last 30 years. Currently, 50% of the population habitually practices some type of sport on a regular basis. This increase has been especially important among women. Women have gone from a passive role in the practice of sports to an absolutely active one. Within this increase in the practice and impact of women's sports, soccer has been the main protagonist. Over the last 20 years, the increase has been spectacular, especially in European countries such as Germany, Holland, Norway and England. Although Spain does not have as many licenses and clubs, both at the amateur and professional level, women's soccer in Spain has also become a trend because of a significant increase both in its practice and in the repercussion at professional level—of the Spanish Women's Football Team itself and of the professional clubs. Therefore, the choice of women's soccer as the specific sport of the study was not accidental.

On the other hand, the development of a business idea based on the binomial between sports and sustainable tourism as a generator of wealth has already been dealt with in multiple studies. The novelty of this work is that it incorporates women's sports into the equation, favoring integration, equality and the dissemination of a sporting practice such as soccer, which is a basically male sport. The increase in the repercussion of women's soccer in order to achieve an increase in its impact, both from a social and economic point of view, requires actions to increase its visibility. Currently, it is private companies that support women's sports in Spain, through different agreements and sponsorship agreements that are either directly to the athletes themselves or through the different federations. One way to go further in this relationship is to link this development to successful business models, and not only to sponsorship systems, as is reflected in this paper.

Finally, the relationship between the generation of opportunities linked to the tourism field was not accidental either. Spain is considered to be one of the preferred tourist destinations by the citizens of the European Union. In 2017, it was the country that received the highest net tourist income, and over the last 5 years, it has been the country where tourist arrivals have increased the most. However, tourism in beach areas and islands, as is the case of Mallorca, is characterized by promoting a tourism model with a summer seasonality that is unsustainable in the long term. The present work allows for the diversification of supply and the search for new markets to reduce the seasonality of the tourism market and improve the economic profitability of the destinations.

All of the above, together with the infrastructures and services that exist in a traditional tourist destination such as Mallorca, and the scarce supply related to the

proposed model, the promotion of the destination as a receiver of women's soccer teams more than justify the study carried out. In short:

- This study proposed a wealth-generating model that represents an estimated income of around 31,000 euros for each women's soccer club that opts for this model from the point of view of the organizing entity;
- It is a model that promotes the development of a sustainable and deseasonalized tourism that takes advantage of existing infrastructures. This point is crucial in tourist destinations linked mainly to the summer period and the so-called sun and beach effect;
- The novelty of the proposed work is that it looks for alternatives to the traditional contributions of existing sports tourism that are typically based on events such as golf, water sports, etc.;
- It is a model that allows for the promotion and increased impact of women's sports in general, and women's soccer in particular. Undoubtedly, these kinds of projects are the ones that contribute to equality and integration, and both the media and economic impact of men's and women's sports.

In other words, apart from the financial indicators necessary in the case of an entrepreneurship project (without this part, the project would not be viable), the work proposed produces synergies in different areas that enhance the value of the model. The following synergies are produced:

- In the field of gender equality: promoting a sport such as women's football, which currently has a great difference in terms of repercussions and economic possibilities compared to men's football;
- In the area of sustainability: generating a business model based on tourism, but with the peculiarity of avoiding overcrowding in the high season with a different development model to the traditional sun and beach model, which leaves little room to maneuver, especially in the summer season;
- In the economic sphere: generating wealth for all the agents involved (transport agencies, hotels, restauration, etc.) and promoting an entrepreneurial project that can be replicated not only in Mallorca, but also in similar tourist destinations.

The limitations of this model are determined by the methodology itself: an entrepreneurial model based on a business plan. Unfortunately, despite being sufficiently worked and referenced, the success of a business model is not based on theoretical aspects but on its practical application. Logically, the problems that may arise in order to achieve the proposed financial results can come from many different areas, from problems arising in the entrepreneurial team (disparity of criteria, errors in decision-making, lack of commitment to the project, etc.) to difficulties from outside the project (problems arising with public administrations, changes in market trends, the appearance of competitors in other geographical areas with similar conditions, etc.). A clear example of this reality is the appearance of the COVID-19 pandemic at the beginning of 2020, which would have made it impossible to offer this service because of health restrictions.

However, these limitations are inherent to any business project and this does not mean that the proposed model cannot and should not be carried out successfully. Therefore, the proposed work incorporates the necessary aspects to be taken into account in future academic references and research related to this academic work. Extensions to the present work should include the development of a public-private consortium that defines a concrete axis of action to favor entrepreneurship models similar to the one proposed. This should include the management of infrastructures and institutional promotion as part of the public communication strategy for tourism and extend the analysis of results to include the overall impact that this sector could bring to the local economy in the field of GDP, increased employment, fiscal resources, etc. Logically, another limitation is due to the timing of this study, in the midst of the health

pandemic. This model can provide a very good opportunity to meet all the objectives set out once the pandemic has been controlled or overcome.

**Author Contributions:** Conceptualization, J.J.V.; Investigation, M.Á.E.F.; Methodology, R.R.F.; Project administration, N.V.M.; Resources, S.N.A.; Writing – review & editing, R.R.F. All authors have read and agreed to the published version of the manuscript.

**Funding:** This research received no external funding.

**Institutional Review Board Statement:** Not applicable.

**Informed Consent Statement:** Not applicable.

**Data Availability Statement:** Consulted and in accordance with the data availability information suggested in the "MDPI Research Data Policies" section at https://www.mdpi.com/ethics (accessed on 1 March 2021).

**Conflicts of Interest:** The authors declare no conflict of interest

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
