# Peer review of "Entrepreneurship, Sport, Sustainability and Integration: A Business Model in the Low-Season Tourism Sector"

_socsci, doi:10.3390/socsci10040117_

Round 1

Reviewer 1 Report

Dear Authors,

I really enjoyed reading your paper and I think that it gives a valuable insight over the possibility of surmounting the shortcomings brought by the COVID-19 pandemics through a combination of tourism and sport activities.

The paper has an overall good structure and I only have a few comments and remarks meant to improve its addressability to the readers:

- please check phrasing and English, as for example:

  • Line 43 – plural “mens” does not exist in English
  • Line 46 – “people are becoming more and more aware of their equation …” – I do not understand the use of the word “equation” here
  • Lines 262-269 han quedado en Espanol
  • Line 299 – 13.3 percentage points instead of 13.3% and 4.9 percentage points instead of 4.9%
  • Lines 461-465 – ambiguous phrasing – lack of sense
  • Lines 659-663 – repeated information
  • Line 701 – Majorca
  • Line 757 – “It helps … island of Mallorca” – poor use of English
  • Lines 806-809 – percentages reduced “a”
  • Line 825 – “british and germans clients” – British and German clients
  • Line 837 – “It will be organized … destination countries” – again not very English
  • Line 847 – “Thus, will be try that the project….” – not English

- In order to make the business model that you proposed replicable, I would suggest better explaining the numbers that you used so as to determine the costs and profit of your model (Tables 11 to 15), as I have found it difficult to understand why you constructed your model based on these figures and not others.

 I look forward to seeing the revised manuscript!

Author Response

Esteemed reviewer,

thank you very much for your contributions to help improve this paper. I hope that the review has met your expectations and that we can continue working on this article. I try to respond to the issues raised:

  • All requested corrections concerning English have been checked (line 43, 46, 262-269, 299, 461-465, 659-663, 701, 757, 806-809, 825, 837 and 847).
  • A section entitled "Literature Review" (lines 620-715) has been included. In this part, an attempt has been made to review new academic contributions, with special emphasis on the most recent ones.  Similarly, examples of similar projects currently in operation have been presented and the dependent variables to be presented in the financial study have been defined. All this is included with further explanations under the section "Results" to try to give more consistency to the proposed data. In the same "Results" section, a final part has been included in which the results obtained are evaluated and contrasted with other studies, indicating their limitations (lines 1069-1080).

On the other hand, an attempt has been made to improve the final content of the work by making the following improvements:

  • The epigraph "Introduction" (lines 30-65) has been improved in order to present from the outset the general structure of the work, the objectives pursued, the methodological approach and the foreseeable conclusions. With this contribution, we hope to make the work more comprehensible from the outset.
    Likewise, at the end of the section "Introduction" (lines 169-187) we have tried to establish the guidelines that the work will follow once the context has been established.
  • The section "Conclusions" has been restructured. The conclusions reached by the work on the basis of the objectives set have been introduced more clearly (lines 1082-1094, lines 1123-1132). In the same "Conclusions" section, a final part has been introduced in which the limitations of the present study and the possible lines of research to be followed are set out (lines 1206-1231).

All updates have been highlighted in yellow to make them easier to find.

Thank you very much for the effort taken in the review. We hope we have met your expectations.

Reviewer 2 Report

This work raises an innovative question that is rarely studied in the traditional literature. However, I believe that the authors repeat the same ideas too many times throughout the work, about what they intend to do with it. On the contrary, there is a need for a more solid and grounded theoretical framework that goes deeper into some of the key ideas that the authors have expressed in their work, such as the existence of similar businesses in the Mediterranean Coast, oriented towards men's football, for example.
The paper should be re-structured so that it is presented in a clearer way: Introduction-Theoretical framework-Methodology-Results-Discussion-Conclusions.
In the conclusions section, the main ideas should be highlighted and the objective of the work should be answered. In addition, it is necessary to add the limitations of the study and the main lines of future research.

Author Response

Esteemed reviewer,

thank you very much for your contributions to help improve this paper. I hope that the review has met your expectations and that we can continue working on this article. I try to respond to the issues raised:

  • A heading called "Literature Review" (lines 620-715) has been included in which the intention is to improve the theoretical framework. In this part, an attempt has been made to review new academic contributions, giving special importance to more recent contributions. In the same way, examples of similar projects currently in operation (mostly related to men's football) have been presented and the dependent variables to be presented in the financial study have been defined. These new academic contributions have tried to relate the objectives set out in the work to the conclusions drawn.
  • The "Introduction" lines 30-65 have been improved in order to present from the outset the general structure of the work, the objectives pursued, the methodological approach and the foreseeable conclusions. New contributions from authors have also been included in this part in an attempt to achieve a more solid theoretical framework.

Likewise, at the end of the "Introduction" section (lines 169-187), an attempt has been made to establish the guidelines that the work will follow once the introductory part has been established.

  • In the "Results" section, a final part has been included in which the results obtained are evaluated and contrasted with other studies, indicating their limitations (lines 1069-1080).
  • Finally the "Conclusions" section has been restructured. The conclusions reached by the work have been introduced more clearly, highlighting the main ideas and relating them to the stated objectives (lines 1082-1094, lines 1123-1132).

In the same "Conclusions" section, a final part has been introduced in which the limitations of the present study and the possible lines of research to follow are set out (lines 1206-1231).

All updates have been highlighted in yellow to make them easier to find.

Thank you very much for the effort taken in the review. We hope we have met your expectations.

Reviewer 3 Report

I congratulate the authors for the article presented for review. I recognize the merit of reconciling noble areas such as the sustainable development of a region through entrepreneurship initiatives in the area of tourism, through the promotion of women's sport.

However, in my opinion, the article needs a qualitative increase in order to be published as a scientific article. My review follows.

Title:

- Is the mention of times of crisis the most appropriate? I think that after reading the article, the authors focus more on the presentation of proposals aimed at combating low seasons (term related to the hotel occupation), than on the presentation of specific suggestions for the crisis situation. I suggest changing to: Entrepreneurship, sport, sustainability and integration: a business model in the tourism sector in low seasons.

Abstract:

- There are more than 200 words - see guidelines;

- Starts with mention of the effects of the pandemic crisis. However, I no longer find references to this specific context throughout the article.

- Later, when analyzing the results and the tables presented, I find that the study is based on information well before the pandemic. They have to review this point to give another conformity to the work.

Keyword:

- I suggest splitting the expression "sustainable and deseasonalized tourism in times of crisis" into (Sustainable tourism); (deseasonalized tourism);

Introduction:

- Between lines 66 and 75 there are study objectives that would be better positioned at the end of the introduction. Thus, I suggest that the authors proceed with a reformulation of the structure of the introduction, which allows a more fluent reading by the readers;

- They should frame the problem, the objectives, the hypotheses under study and the expected conclusions;

- This section should end with the presentation of the structure that the rest of the article follows. Again, this helps to situate the reader;

- 9 of the 14 quotes found here are over a decade old;

- Several considerations need to be accompanied by the source.

Literature review:

- Line 242 - which study do the authors refer to?;

- Translate the paragraph between lines 262 and 269 into English;

- It was important that at the end of the literature review, the research model followed (image with the connection and relationship of the variables) was presented and the study hypotheses presented, which will be discussed in the next section of the results. This too would serve to guide and focus the reading of the article;

- Once again, several statements, findings or definitions need a theoretical framework, they must mention the sources;

Results:

- The first 4 paragraphs of this section still seem to be part of the literature review;

- The discussion of results requires a deeper confrontation with the literature. You must demonstrate whether the results obtained in your study are in line or not, which has been written by other authors. This point is important and should deserve great attention from the authors;

Conclusion:

- It is important for the authors to state clearly what are the contributions of this study to the theory and what limitations they recognize to the study.

Other aspects to take into account:

Tables: see formatting according to guidelines;

Citations are not applied correctly - see guidelines;

The spacing between paragraphs and the alignments at the beginning of the sentences need to be revised - I recommend formatting according to guidelines.

Finally, I wish the authors good luck and see these suggestions as an opportunity to improve their work.

Author Response

Esteemed reviewer,

thank you very much for your contributions to help improve this paper. We hope that the review has met your expectations and that we can continue working on this article. We try to respond to the issues raised:

  • The title has been changed to put more emphasis on the objective of the work directed at combating low seasons in tourism.
  • The abstract has been restructured so that it is no longer than 200 words. Similarly, references to the health crisis have been omitted from the abstract in order to reflect the objectives of the work in a different way.
  • The key words have been redefined by omitting the reference to the health crisis.
  • The location of part of the introduction has been changed (former lines 66-75, in the new paper 169-177).
  • The epigraph "Introduction" (lines 30-65) has been restructured to present from the outset the general structure of the work, the objectives pursued, the methodological approach, the study hypotheses and the expected conclusions. New contributions from authors have also been included in this part in an attempt to achieve a more solid theoretical framework. With this new approach, it is hoped that the reader will have a better overall understanding of the work from the outset.
  • Likewise, at the end of the "Introduction" section (lines 169-187), an attempt has been made to define the structure that the rest of the work will follow.
  • A section entitled "Literature Review" (lines 620-715) has been included in order to improve the theoretical framework. In this part, an attempt has been made to review new academic contributions, giving special importance to more recent contributions. These new academic contributions have tried to relate the objectives set out in the work with the conclusions provided. In the restructured part of the "Introduction" we have also tried to review the literature of the work with more recent contributions from authors.
  • An attempt has been made to explain the source of the considerations made in the paper (lines 46, 238:241, 283, 501 to give some examples).
  • The references in the previous paper have been corrected (line 242 of the previous document) and the paragraph that was in Spanish has been translated into English (lines 301-307).
  • At the end of the "Literature Reviem" section, examples of similar projects currently in operation (mostly related to men's football) have been given and the dependent variables to be presented in the financial study have been defined. In this way, an attempt has been made to explain the research model followed with the study hypothesis.
  • The first paragraphs that were located in the "Results" section have been included in the "Literature Review" section (lines 620-641).
  • A final part has been included in the "Results" section (lines 745-755, 763-767) in which an attempt has been made to explain the use of the variables used with the results obtained. Similarly, a final part has been included (lines 1069-1080) in which the limitations raised in this study are discussed and an attempt is made to relate them to similar limitations in studies related to entrepreneurship projects.
  • The "Conclusions" section has been restructured. The contributions achieved by the work have been introduced in a clearer way, highlighting the main ideas and relating them to the stated objectives (lines 1082-1094, lines 1123-1132).
    In the same "Conclusions" section, a final part has been introduced in which the limitations of the present study and the possible lines of research to follow are set out (lines 1206-1231).
  • Finally, an attempt has been made to respond to the guidelines set out regarding paragraph spacing, alignments, citations and tables.

All updates have been highlighted in yellow to make them easier to find.

Thank you very much for the effort taken in the review. We hope we have met your expectations.

Round 2

Reviewer 3 Report

Dear authors,

Thank you for taking the suggestions into consideration. Congratulations on the reformulation made to the initial version of the article. I believe that there has been a qualitative leap that will allow its publication. Good luck for future work.